# Information feedback in relative grading: Evidence from a field experiment

**Shinya Kajitani** [1]◐, **Keiichi Morimoto**[2]◐, **Shiba Suzuki**[3]◐*

**1** Faculty of Economics, Kyoto Sangyo University, Kyoto-city, Kyoto, Japan, **2** School of Political Science and Economics, Meiji University, Chiyoda-ku, Tokyo, Japan, **3** Faculty of Economics, Seikei University, Musashino-city, Tokyo, Japan

◐ These authors contributed equally to this work.
* shiba.suzuki@econ.seikei.ac.jp

**Data Availability Statement:** All relevant data are within the paper and its Supporting Information files.

**Funding:** The author(s) received no specific funding for this work.

## Abstract

Previous studies have revealed the role of relative performance information feedback on providing agent incentives under a relative rewarding scheme through laboratory experiments. This study examines the impact of relative performance information feedback of students' performance on their examination score under the relative grading scheme in an actual educational environment. Conducting a randomized controlled trial in a compulsory subject at a Japanese university, we show that the relative performance information feedback has a significantly positive impact on the students' examination score *on average*, but that the average positive impact is derived by the improvement of low-performing students.

## Introduction

Does relative performance information feedback improve a student's incentive to study under a relative grading scheme? Many consider information feedback associated with a reward environment as an efficient way of increasing the incentives of students to study. "Relative grading" or "grading on a curve" is widely used in grading students. Conducting a randomized controlled trial in a compulsory subject required for university graduation, we examine the impact of relative performance information feedback on students' examination scores.

In relative grading, a student's grade depends on her position in the class score distribution. To understand student incentives in a relative grading scheme, Becker and Rosen [1] extend the rank-order tournament model of Lazear and Rosen [2] and emphasize that the student's learning effort depends on her position in the distribution of academic attainment. Andreoni and Brownback [3] construct a theoretical model of relative grading employing an all-pay auction and demonstrate that low skilled subjects decrease effort but high skilled subjects increase effort as the auction size increases. These suggest that relative performance information feedback affects student decision-making in providing effort. Besides, in actual schooling environments, multiple examinations typically grade students. Aoyagi [4] and Ederer [5] theoretically analyze information feedback in a dynamic tournament context. It is then worth considering the relationship between information on a student's relative position in the distribution of

**Competing interests:** The authors have declared that no competing interests exist.

earlier examination scores and her incentive to provide study effort for the following examination in actual schooling environments.

How then does this relative performance information feedback affect the students' incentive to study under a relative grading scheme in a multiple examinations environment? In a relative grading scheme, to obtain a better grade a student needs to receive a higher score than her opponents do. That is, an opponent's score serves as a threshold she must exceed. In this grading environment, the relative performance information feedback is then a signal of the effort she should provide. For example, when relative performance information feedback tells the student that her current score is relatively low, she understands that she has to provide a higher level of effort to rise above the threshold. Conversely, she may give up, saving the cost of effort.

Some laboratory experiments reveal the role of relative performance information feedback with respect to relative rewarding. These studies generally suggest that there is no guarantee that the relative performance information feedback has a positive impact on the students' incentive to study. For example, Eriksson et al. [6] and Freeman and Gelber [7] conclude that relative performance information feedback lowers the performance of subjects whose interim performance is relatively low. However, those subjects whose midterm performance is relatively high do not slacken off. In contrast, Ludwig and Lünser [8] examine the effects of effort information in a two-stage rank-order tournament. They demonstrate that laboratory subjects who lead tend to lower their effort, but those who lag increase it relative to the first stage, while the subjects who lead exert a greater effort than those who lag. Thus, the impact of relative performance information feedback may vary according to the initial level of attainment.

In an actual educational environment, previous studies focus on the impact of relative performance information feedback on student incentives under absolute grading. For example, Azmat and Iriberri [9], using data from Spanish high schools, and Tran and Zeckhauser [10], in a field experiment of Vietnamese university students, demonstrate that relative performance information feedback raises the performance of students when rewarded absolutely. Both these studies argue that if students have competitive preferences, which means that they inherently prefer receiving a higher rank than others, relative performance information has a positive impact on their incentive to study on average.

The question in this paper is whether relative performance information feedback improves student examination scores in an actual relative grading environment where students sit for examinations on multiple occasions. To examine this issue, we conduct a field randomized controlled trial employing the compulsory subject of economics at a Japanese university. In this course, after students sat two examinations (the midterm examination and the final examination), instructors calculated the students' final raw scores mainly by taking the weighted average of their two examination scores. However, the students' grades were evaluated by grading on a curve, with the instructors adjusting the students' final raw scores subject to the entire final raw score distribution to obtain the reasonable pass rate. In our experiment, we allocated more than 200 students into a control group and a treatment group immediately following the midterm examination. We only provided students in the treatment group with feedback on their midterm examination relative performance and explored the impact of this feedback on student performance in the final examination.

This study is the attempt to investigate the impact of relative performance information feedback on student incentives to study in an actual educational environment encompassing relative grading. We show the significant positive impact of relative performance information feedback on the students' final examination scores *on average*. Note that because students cannot graduate from the university unless they receive credit in this subject, they care about whether they can receive credit. In other words, the threshold between a pass and a fail in the course is significant for students. Moreover, the threshold depends not only on the students'

ranks in the distribution of the final raw scores but also on their final raw scores *per se*. By considering the threshold, we demonstrate that the average positive impact on the final examination scores is through the improvement of low-performing students in the midterm examination.

The remainder of the paper is organized as follows. Section "" describes the experimental design. Section "Balance between the control and treatment groups" presents the empirical framework and reports the estimation results and discusses the findings. Section "Discussion" concludes.

## Materials and methods

This experiment was approved by Meisei University's research ethics committee on the Use of Human Subjects (Application No. H26-002). Before conducting the studies, we obtained informed consent from all subjects.

### Description of the randomized trial

This section provides details of the randomized trial, performed using first-year students in an economics department at a Japanese private university. We begin by describing the flow of interventions in the experiments, which are displayed in Fig 1. The academic year comprised first and second semesters: the first semester began in April 2012 and ended in July 2012; the second semester began in September 2012 and ended in January 2013. We conducted a mathematical achievement test (referred to as the Pretest of Mathematics) immediately following university entrance. Students enrolled in two compulsory introductory economics courses in their first year: Economics I in the first semester and Economics II in the second semester. In Economics I and II, we administered midterm and final examinations to grade students. While the midterm and final examinations in Economics I were in May and July 2012, those in Economics II were in November 2012 and January 2013. We note that the score for the Pretest of Mathematics was independent of the grades for Economics I and II. The dotted vertical lines in Fig 1 represent the timing of the examinations.

We evaluated the students in both Economics I and II using the same grading scheme, namely, grading on a curve. The instructors explained this grading scheme in detail to the students in Economics II at the beginning of the second semester. We provide details of the grading on a curve scheme later.

We divided students into four classes. According to their score in the Pretest of Mathematics, we placed all students with a top-40 score in one small class. Hereafter, we refer to this as Classroom 1. The designation of Classroom 1 is for purely educational purposes. For example, in teaching economics, we used a different level of mathematics in Classroom 1 and the other classrooms. We then randomly allocated the remaining students to the other three classes. Hereafter, we refer to these as Classrooms 2, 3 and 4. We fixed all class enrollments and instructors across both semesters. While each class had its own instructors, such that all classes were held in the third period (12:55 p.m.–14:25 p.m.) on Wednesday, all students took the same examination using a multiple-choice computer-scored answer sheet at the same time.

The experimental intervention was implemented immediately after the midterm examination in Economics II, which then randomly assigned all students to the treatment or control group. In the first class time after the midterm examination, we handed students letters revealing their score for the midterm examination. In addition, the letters given to students in the treatment group also reported their ranks in the midterm examination. We did not include this information in the letters to the students in the control group. The student letter content is similar to that used by Ashraf et al. [11]. Figs 2 and 3 reproduce the information provided to

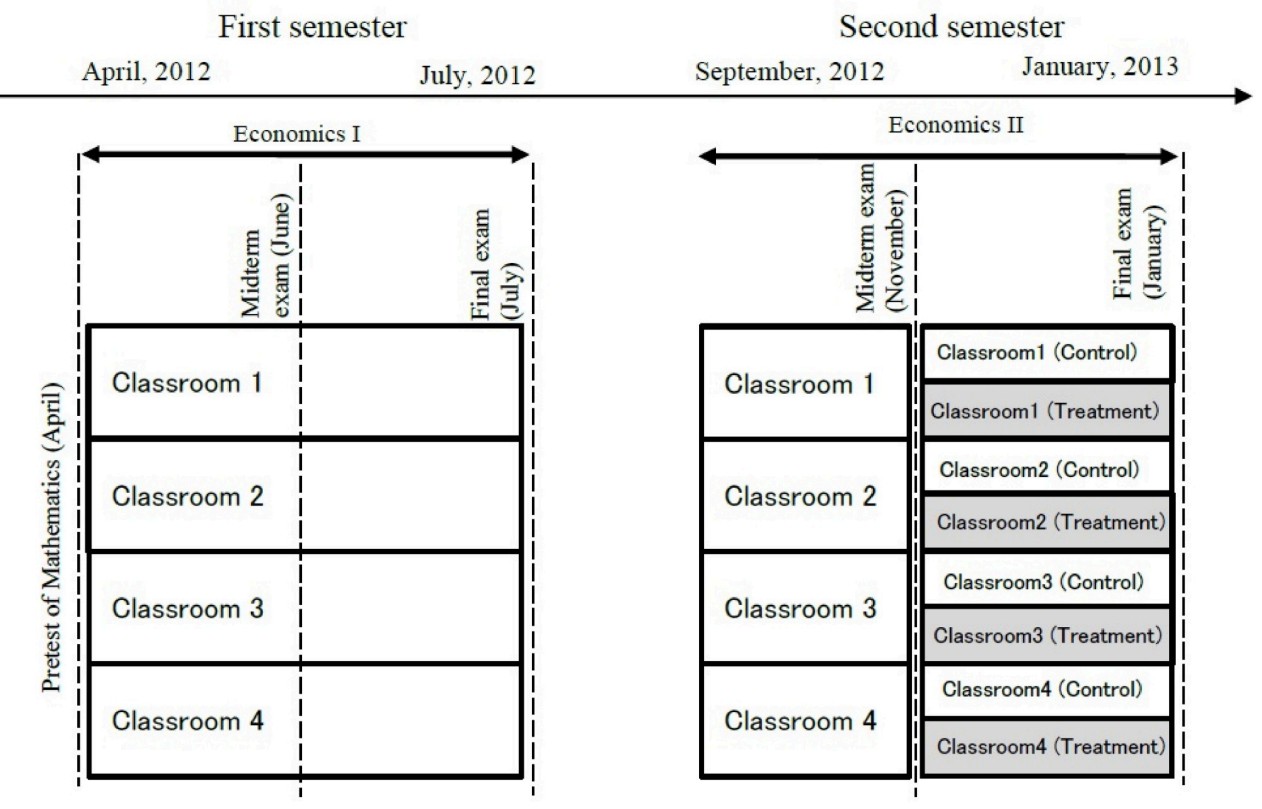

**Fig 1. The flow of interventions in the experiment.**

the students in the treatment and control groups. On this basis, while students in the treatment group knew their precise rank, students in the control group would only have a vague awareness.

There are several points to note in our randomized control trial. First, we exclude some students who did not receive the letter regarding the midterm examination from our sample. Because some students were absent from the class time just after the midterm examination, we could not hand letters to these students. Therefore, we do not consider these students as subjects in our experiment.

Second, our experimental design cannot exclude the possibility that some students may have exchanged their rank information. Because our experimental design is similar to that of Tran and Zeckhauser [10] save the grading scheme, we share the same problem that students in both the control and treatment groups sit in the same classroom, making the exchange of rankings a genuine possibility. However, it would be generally difficult for a student in the control group to identify a student in the treatment group with exactly the same score; students in the treatment group know their exact rank, while students in the control group do not. Therefore, the impact of relative performance information feedback, if any, is captured by our experimental design. We discuss this further in Subsection "Balance between the control and treatment groups" and Section "Balance between the control and treatment groups."

Third, instructors were not blind to the treatment a certain student had been assigned because the letter was not put in an envelope and instructors could confirm this information. However, aside from handing letters to students, instructors could not confirm which students

Introductory Economics, Second Semester, 2012
A report on the result of your midterm examination

Student ID          **7 digit ID**          Name          **Name of the student**
Classroom          **Instructor's name**

Your score in the midterm examination is                    **49**/110
Of them, your score in problems of mathematics is          **6**/10

Within four classrooms, you are **146**th out of 285 students.

Original (written in Japanese)

2012 年度経済学通論 2　中間テスト結果

学籍番号　12E1-XXX　　氏名　XXXX
経済学通論 2 クラス（XXXX）

中間テストの点数　　　　　49 点／110 点満点
うち　数学出題分　　　　　6 点／10 点満点

学年全体でのあなたの順位　　　285 人中　146 位

**Fig 2. The letter to students (treatment group).**

were assigned to treatment or control, and it would be difficult for instructors to remember this information. In addition, class attendance or participation, such as the number of times a student spoke in class, was not evaluated at all. Thus, whether instructors were blind to which students were assigned to treatment or control would have little impact on the experimental results.

Finally, students are exogenously assigned in the treatment group and the control group in Economics II. The class assignment is neutral for our randomized control trial. The reason why we placed students with a top-40 score in the Pretest of Mathematics in Classroom 1 is purely for educational purposes. A substantial mathematical background is generally important to understand economics. However, there was a disparity in the mathematical background of freshman students, and many did not have sufficient knowledge of mathematics. In Economics I, to provide these students with remedial education in mathematics, we placed them in Classroom 2, 3, or 4. Consequently, we placed few students who had sufficient mathematical background in Classroom 1. Nevertheless, except for the remedial education in mathematics, the course material and examinations in economics were the same across Classrooms 1–4 in Economics I. In Economics II, the course material and examinations in economics were also identical, and no remedial education in mathematics in any of the classrooms was given.

> Introductory Economics, Second Semester, 2012
> A report on the result of your midterm examination
>
> Student ID        **7 digit ID**        Name    **Name of the student**
> Classroom        **Instructor's name**
>
> Your score in the midterm examination is                    **49**/110
> Of them, your score in problems of mathematics is           **6**/10

Original (written in Japanese)

> 2012年度経済学通論2　中間テスト結果
>
> 学籍番号　12E1-XXX　　氏名　XXXX
> 経済学通論2クラス（XXXX）
>
> 中間テストの点数　　　　　49点／110点満点
> うち　数学出題分　　　　　6点／10点満点

**Fig 3. The letter to students (control group).**

Instead, students are required to understand the basic concepts in economics in Economics II. Most of questions in examinations in Economics II are not mathematical but of a multiple-choice type. Therefore, initial mathematical background itself has less of an impact on achievement in Economics II.

## The grading scheme

In Economics II, a final raw score was calculated as follows: perfect raw scores were 110 points, in which 100 points were for the two examinations (the midterm and final examinations) and the remaining 10 points for the number of homework submissions. The examination score of 100 is divided into "40% of the midterm examination score" *and* "60% of the final examination score." "The number of homework submissions" is based on 10 homework assignments, each worth one point.

It should be noted that the instructors can adjust students' final raw score upward considering the entire final raw score distribution. This upward adjustment introduces uncertainty into threshold scores between one grade and another, especially the threshold score between pass and fail. The official university's guidelines recommend that instructors should assign the

## ■ Grading

In the general courses, students are graded by 5 categories, S/A/B/C (pass) and F (fail).
In *Seminar for freshman I and II,* students are graded by 2 categories P (pass) and H (fail).

\

| Scores | Grade | Pass/Fail |
|--------|-------|-----------|
| 100−90 | S | Pass |
| 89−80 | A | |
| 79−70 | B | |
| 69−60 | C | |
| 59− 0 | F | Fail |

**Fig 4. The slide to explain the grading scheme (originally written in Japanese).**

first grade (S) to scores 90 and over, the second grade (A) to scores between 80 and 89, the third grade (B) to scores between 70 and 79, the fourth grade (C) to scores between 60 and 69, and fail (F) to scores below 60. A student who marks F fails the course. These grading criteria were explained in the guidance for freshman students in April. Fig 4 shows the slide that was used for this purpose. Therefore, students learnt that the official guideline related the scores to the grading criteria. At the same time, in this guidance, instructors also verbally announced that there was a possibility that raw scores would be adjusted upward to obtain a reasonable pass rate. That is, when the instructors give grades to the students afterward, they can adjust students' final raw scores upward depending on the entire final raw score distribution. Meanwhile, none of the students ever knows whether the instructors will adjust their final raw scores in advance of taking the final examination. For example, because the average final raw score in Economics I was quite low (52.7 points out of the perfect score of 100), the instructors decided to add 9 points to all the students' final raw scores. This upward adjustment made the student whose final raw score was above 50 receive a credit for Economics I. In contrast, because the instructors did not finally adjust the students' final raw score in Economics II, only the students whose final raw score was 60 and over got credit for Economics II.

There are three points to note in our grading scheme. First, students in Economics II knew the grading scheme described above. The instructors had already explained this grading scheme in detail at the beginning of the second semester, in addition to the guidance given in April. Moreover, this grading scheme had already been employed in Economics I, in which students were graded based on scores in the midterm examination (June) and final examination (July). After the midterm examination, instructors handed a letter in person to the students to inform them of their own absolute score, the average score of all students, and the class average score. Students also received the official grade report from the university in August. This grade report was also available online. Therefore, in Economics I, students could

decide how much effort to put into the final examination after they learnt their absolute scores in the midterm examination; after that, they also learnt whether they had passed the course or not. This flow of events was the same in Economics II. Thus, students in Economics II had already experienced the same grading scheme in Economics I. It can be considered that students understood the relationship between effort input in the midterm and final examinations and their grade. In other words, Economics I served as a practice session that familiarized students with the grading scheme used in Economics II.

Second, instructors grade all students in the four classrooms using the same grading criteria in Economics II. All students had to register for the same courses of Economics I and II. After registration, instructors assigned Classrooms 1, 2, 3, and 4 to students. As noted, all students took the same examination at the same time using a multiple-choice computer-scored answer sheet. It was announced several times during the course that the grading criteria were the same for all four classes. We therefore believe that students considered that they were graded according to the same criteria across the four classes. In addition, four instructors decided the cutoff scores after consultation, and one instructor registered the grades of all the students on behalf of the other three instructors. Instructors could not deviate from the agreed pass scores. Therefore, students compete not only with students in their classroom but also the other classrooms; whether students pass or fail will depend on their relative position in the entire score distribution of more than 200 students. The students in Economics II were exposed to the uncertainty of the threshold they must exceed to pass the course.

Finally, in our experiment, when the instructors gave grades to the students afterward, they may well adjust the students' final raw scores upward but never adjusted their final raw scores downward. Under these circumstances, a student whose final raw score was over 60 points got credit for Economics II. If the student who already got 60 points in the midterm examination gets at least 60 points in the final examination, she can get credit for Economics II. That is, whether students pass or fail depends not only on the students' rank in the distribution of the final raw scores but also on their final raw scores *per se*.

## Balance between the control and treatment groups

Table 1 provides the total number of students and the means and standard deviations of the midterm examination scores in Economics II for the control and treatment groups. Table 1 also shows how we randomly divided these students into the control and treatment groups.

In total, 284 students took midterm examinations, and their mean score was 49.57. We randomly divided these students into control and treatment groups. However, some students failed to receive the letter. Consequently, in our experiment, there are 255 subjects with a mean score of 50.67. There were 130 and 125 students in the control and treatment groups, respectively. The mean scores for the control and treatment groups are 51.48 and 49.82, respectively, and there is no significant difference in the mean scores between the control and treatment groups, as shown in row (a) in Panel B.

One point to note is the differences between classrooms. Table 1 also shows that we randomly divided students into the control and treatment groups if we consider these differences. The mean score in the midterm examination in Classroom 1 is much higher than that in Classrooms 2–4 because we enrolled students with a top-40 mark in the Pretest of Mathematics in Classroom 1. The number of students who received the letter in Classrooms 2–4 is 215, while that in Classroom 1 is 40. The mean for students who received the letter in Classroom 1 is 65.48 and that in Classrooms 2–4 is 47.92. There is no significant difference in the mean scores between the control and treatment groups across Classrooms 2–4. While the number of students and the mean for the control group is 106 and 48.21, respectively, those for the treatment

**Table 1. Randomization checks.**

Panel A. Descriptive statistics of the midterm examination scores.

| | | 1 | 2 | 3 | 4 | 5 |
|---|---|---|---|---|---|---|
| | | Classrooms 1–4 | Classrooms 2–4 | Classroom 1 | Male | Female |
| I. All | Obs. | 284 | 244 | 40 | 248 | 36 |
| | Mean | 49.57 | 46.96 | 65.48 | 49.40 | 50.72 |
| | S.D. | 17.36 | 16.24 | 15.54 | 17.60 | 15.72 |
| II. Receive a letter | Obs. | 255 | 215 | 40 | 221 | 34 |
| | Mean | 50.67 | 47.92 | 65.48 | 50.77 | 50.03 |
| | S.D. | 17.02 | 15.85 | 15.54 | 17.22 | 15.90 |
| -i. Control | Obs. | 130 | 106 | 24 | 113 | 17 |
| | Mean | 51.48 | 48.21 | 65.96 | 51.94 | 48.47 |
| | S.D. | 18.64 | 17.70 | 15.84 | 18.88 | 17.23 |
| -ii. Treatment | Obs. | 125 | 109 | 16 | 108 | 17 |
| | Mean | 49.82 | 47.63 | 64.75 | 49.55 | 51.59 |
| | S.D. | 15.18 | 13.90 | 15.56 | 15.29 | 14.82 |

Panel B. Mean-comparison test (Welch t-test).

| | t-value | P-value |
|---|---|---|
| (a) Comparison (1, II-i) with (1, II-ii) | 0.781 | 0.435 |
| (b) Comparison (2, II-i) with (2, II-ii) | 0.264 | 0.792 |
| (c) Comparison (3, II-i) with (3, II-ii) | 0.239 | 0.813 |
| (d) Comparison (4, II-i) with (4, II-ii) | 1.037 | 0.301 |
| (e) Comparison (5, II-i) with (5, II-ii) | -0.566 | 0.576 |

group are 109 and 47.63, respectively. We do not reject the null hypothesis that "the mean values of the two groups are not different," as shown in row (b) in Panel B. In addition, as for Classroom 1, the number of students for the control and treatment groups are 24 and 16, respectively, and the mean scores for the control and treatment groups are 65.96 and 64.75, respectively. We again do not reject the null hypothesis that "the mean values of the two groups are not different," as shown in row (c) in Panel B. There is also no significant difference in the mean scores between the control and treatment groups by sex (rows (d) and (e) in Panel B).

Another point to note is the possibility of information spillover. Because Classroom 1 contains only 40 students, one may wonder if students in the control group can know their rank by communicating with students in the treatment group. However, such communication and potential spillover should be minimal because the information set in Classroom 1 is sparse. Only seven students in the control group can find a student in the treatment group with exactly the same score. The remaining 17 students in the control group cannot find such a student, meaning that these students can only know a range of rank but cannot know their precise rank. For example, there are two students in treatment group: the score of one of them is 55 and the rank of her is 95, and the score of the other of them is 52 and the rank of her is 120. Also, there is one student in the control group whose midterm score is 53. However, there is no student in the treatment group whose midterm score is exactly 53. In this case, the student cannot know her own rank accurately. All she can know is that her rank is between 96 and 119. Of course, because there is no guarantee that she will successfully find students in the treatment group with scores of 55 and 52, her estimation about her own rank could be more

ambiguous. Therefore, our experimental procedure could capture the effect of the difference in the degree of uncertainty in the rank perception on the final exam scores.

## Results and discussion

### The effects of relative performance information feedback

The randomized controlled trial means that we obtain two groups that are statistically equivalent to each other. This study captures the effects of relative performance information feedback on the final examination scores, that is, the average effects of assignment to the treatment group versus assignment to the control group. We simply use OLS to estimate the effects of treatment interventions on test scores (e.g., Levitt et al. [12]) and employ the following empirical framework:

$$Y_{Fi} = \alpha D_i + X_i \beta + \epsilon_i, \tag{1}$$

where $Y_{Fi}$ denotes the scores in the final examination for student $i$. When student $i$ took the midterm examination but not the final examination, we treat $Y_F$ for student $i$ as zero. $D_i$ is a dummy variable equal to one if student $i$ is given information on her relative rank in the midterm examination (i.e., the student is in the treatment group), and zero if student $i$ is not given this information (i.e., the student is in the control group). $X_i$ denotes the covariates including a constant term, and $\epsilon_i$ are disturbances.

Table 2 provides descriptive statistics for all the variables used in this estimation model. In our experiment, we randomly assigned all students to the treatment or the control group using a random number generator. While associated covariates, the midterm examination scores and the classrooms are also randomly selected, we are concerned about ex post differences in the values of the midterm examination scores. We include the midterm examination scores $Y_{Mi}$ for student $i$, dummy variables for students in different classrooms, $Class1_i$, $Class2_i$, and $Class3_i$ (the classroom fixed effects), and a female dummy variable $Female_i$ in the vector $X_i$.

Column (1) in Table 3 report the results of estimating Eq (1). We report robust standard errors that are not clustered by classroom but that are adjusted for individual heterogeneity. In our experiment, there is no cluster in the population of interest not represented in the sample. Moreover, clustered standard errors with only a few clusters (e.g., just four classrooms in our experiment) could not be reliable as explained by Angrist and Pischke [13]. There is no need to adjust standard errors for clustering once fixed effects are included as argued by Abadie et al. [14]. The magnitude of the coefficient for $D$ is 3.517 and statistically significant. This indicates that the scores in the final examination for students who received information on their relative rank in the midterm examination are 3.517 points higher on average than the scores for students who did not get this information. When we use the difference between the midterm examination score and the final examination score ($Y_{Fi} - Y_{Mi}$) as the dependent variable, we see the significantly positive impacts of the relative rank in the midterm examination. As shown in Column (2), the magnitude coefficient for $D_i$ is 3.706. The coefficient of $Class2$ in Column (1) and those of $Class1$–$Class3$ in Column (2) are significant. The classroom fixed effects would absorb instructor fixed effects as well as peer effects and other classroom-level factors. These effects and factors may be associated with test scores.

In terms of other research considerations, such as the experimental design employed by Tran and Zeckhauser [10], we divided students into control and treatment groups within each classroom. Because $D_i$ was randomly assigned, the coefficient for $D_i$ have a causal interpretation. However, the coefficient for $D_i$ tells us the causal effect of the offer of treatment, including the fact that some of those offered have shared their ranks with their classmates, even if it is difficult for a student in the control group to identify a student in the treatment group with

**Table 2. Descriptive statistics.**

| Variable | Definition | Mean | Std. Dev. | Min | Max |
|---|---|---:|---:|---:|---:|
| Total (Obs. = 254) | | | | | |
| $Y_F$ | Score in the final examination | 63.807 | 20.341 | 0 | 100 |
| $D$ | = 1 if student is given information on her relative rank in the midterm examination, = 0 if elsewhere | 0.488 | 0.501 | 0 | 1 |
| $Y_M$ | Score in the midterm examination | 50.602 | 17.020 | 12 | 102 |
| $H$ | = 1 if $Y_{Mi} \geq 60$, = 0 if elsewhere | 0.291 | 0.455 | 0 | 1 |
| Class1 | = 1 if in the classroom 1 (math class), = 0 elsewhere | 0.157 | 0.365 | 0 | 1 |
| Class2 | = 1 if in the classroom 2, = 0 elsewhere | 0.295 | 0.457 | 0 | 1 |
| Class3 | = 1 if in the classroom 3, = 0 elsewhere | 0.264 | 0.442 | 0 | 1 |
| Female | = 1 if student is a female, = 0 elsewhere | 0.130 | 0.337 | 0 | 1 |
| Treatment (Obs. = 124) | | | | | |
| $Y_F$ | | 64.927 | 17.372 | 0 | 92 |
| $D$ | | 1 | 0 | 1 | 1 |
| $Y_M$ | | 49.677 | 15.154 | 12 | 95 |
| $H$ | | 0.250 | 0.435 | 0 | 1 |
| Class1 | | 0.129 | 0.337 | 0 | 1 |
| Class2 | | 0.306 | 0.463 | 0 | 1 |
| Class3 | | 0.266 | 0.444 | 0 | 1 |
| Female | | 0.129 | 0.337 | 0 | 1 |
| Control (Obs. = 130) | | | | | |
| $Y_F$ | | 62.738 | 22.833 | 0 | 100 |
| $D$ | | 0 | 0 | 0 | 0 |
| $Y_M$ | | 51.485 | 18.642 | 18 | 102 |
| $H$ | | 0.331 | 0.472 | 0 | 1 |
| Class1 | | 0.185 | 0.389 | 0 | 1 |
| Class2 | | 0.285 | 0.453 | 0 | 1 |
| Class3 | | 0.262 | 0.441 | 0 | 1 |
| Female | | 0.131 | 0.338 | 0 | 1 |

[1] Because we exclude a student whose midterm score was revised from the sample, our final sample comprised 254 students.

[2] Our experiment was performed using freshman students in an economics department at a private Japanese university. Using the observed standard deviation of 17.02 from the midterm examination scores shown in Table 2, our sample size can detect a 5.5 point treatment effect of relative performance information feedback at 10% significance level with 80% power (the sample size per group required to detect the significant effect is 120). Thus, our sample size is underpowered to estimate effect sizes below this cutoff, many of which could have a positive effect.

precisely the same score. This leads to the suggestion that the coefficient for $D_i$ may be small relative to the average causal effects on those in fact treated.

## The heterogeneity in the effects due to high- or low-performance in the midterm examination

To understand students incentive to study in our experiment, it is useful to identify the causes of our treatment effects. Czibor et al. [15] conduct a field experiment in a Dutch university and compare relative and absolute grading. They find no significant differences in examination scores between a relative and an absolute grading scheme. On this basis, Czibor et al. [15] contend that rank incentives are weak if students adopt a just-pass behavior. That is, if students only care about whether they can pass the course, they will not put in effort to gain a higher

**Table 3. Estimation results: The effects of relative performance information feedback.**

| Dependent variables | $Y_F$ | $Y_F - Y_M$ |
|---|---|---|
| | (1) Coeff. | (2) Coeff. |
| $D$ | 3.517* (2.023) | 3.706* (2.067) |
| $Y_M$ | 0.752*** (0.075) | |
| $Class1$ | -2.775 (3.166) | -7.694*** (2.953) |
| $Class2$ | -4.589* (2.654) | -4.947* (2.644) |
| $Class3$ | -4.552 (2.973) | -5.970** (2.990) |
| $Female$ | 0.978 (3.112) | 1.268 (3.183) |
| $Constant$ | 26.884*** (4.773) | 15.478*** (2.416) |
| Obs. | 254 | 254 |
| Adjusted $R^2$ | 0.38 | 0.03 |
| F test $H_0$: all the coefficients except the constant are jointly zero | 22.59*** | 2.77** |

[1] *, ** and *** indicate statistical significance at the 10%, 5% and 1% levels, respectively.

[2] Standard errors in parentheses are adjusted for heterogeneity.

[3] Because we exclude a student whose midterm examination score was revised from the sample, our final sample comprised 254 students.

rank than that to which they aspired. Even if graded relatively, relative performance information feedback may exert different impacts on student incentives to study depending on their attitude toward obtaining higher grades. Our grading scheme is close to the classroom setting of just-pass students considered by Czibor et al. [15]. That is, although grades consist of S, A, B, C, and F in our experiment, the threshold between C and F is distinguished from the other thresholds. This is because students must pass the course in order to graduate. In addition, in our experiment, when the instructors gave grades to the students afterward, they never adjusted their final raw scores downward. Under these circumstances, the threshold between C and F depends not only on the students' ranks in the distribution of the final raw scores but also on their final raw scores. Therefore, in our experiment, the higher a student's midterm examination score, the lower the required score in the final examination for her to receive credit in Economics II. These indicate that the rank for students with a higher (lower) midterm examination score has less (more) tangible benefits. If so, the relative performance information feedback could affect high- or low-performing students differentially.

We can visually observe the heterogeneity in the effects when depicting a violin plot of the final examination scores by group in Fig 5. When based on below the lower level of the median scores in the final examination, the kernel density for the treatment group is narrower than that for the control group. This suggests that the relative performance information feedback may raise effort for students whose midterm examination scores were relatively lower, while the feedback may decrease effort for students whose midterm examination score were relatively higher. That is, the relative performance information feedback could affect high- or low-performing students differentially.

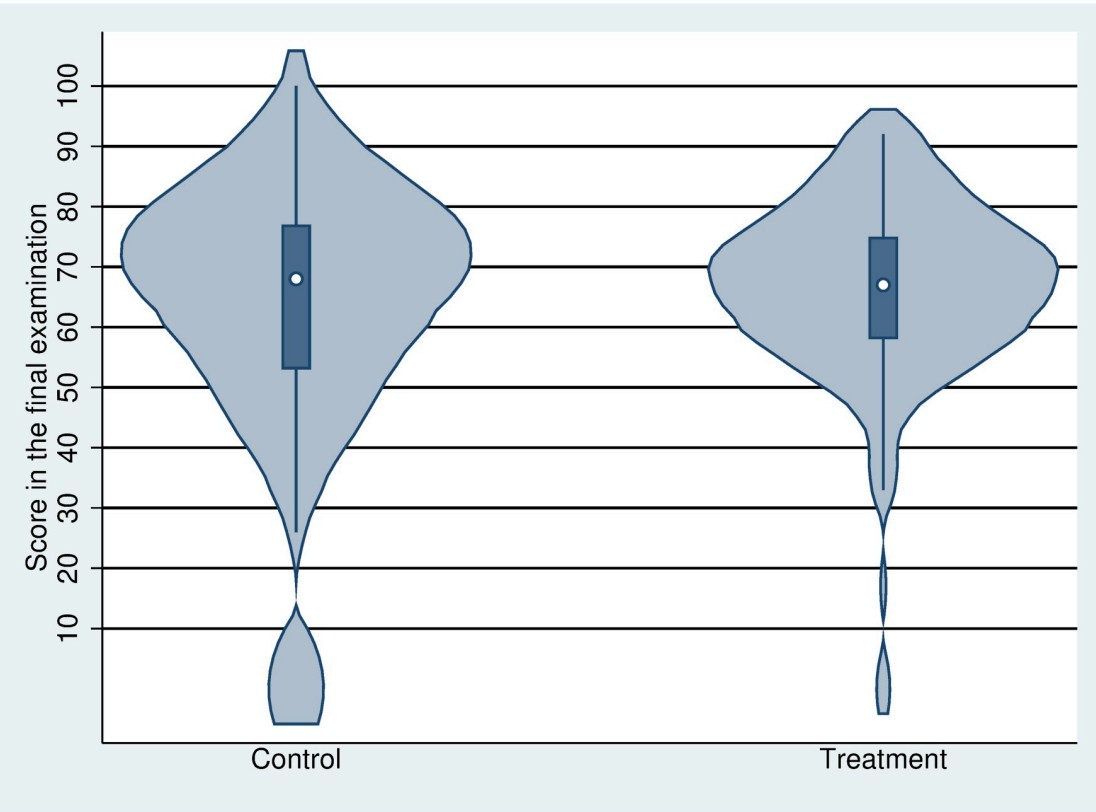

**Fig 5. The violin plot of the final examination scores.** The violin plot comprises a combination of the box plot and the density trace. This includes a marker for the median scores of the final examination, a box indicating the interquartile range and spikes extending to the upper- and lower-adjacent values, and plots of the estimated kernel density.

To examine the heterogeneous effects, we consider the following two equations:

$$Y_{Fi} = \alpha_1 D_i + \alpha_2 (D_i \times Y_{Mmci}) + X_i \beta_1 + u_{1i}, \tag{2}$$

$$Y_{Fi} = \alpha_3 D_i + \alpha_4 (D_i \times H_i) + \alpha_5 H_i + X_i \beta_2 + u_{2i}. \tag{3}$$

In Eq (2), we additionally include $Y_{Mmci}$ instead of $Y_{Mi}$ and the interaction term $D_i \times Y_{Mmci}$ in Eq (1). $Y_{Mmci}$ denotes the midterm examination scores that are centered at the mean of $Y_{Mi}$. On the other hand, in Eq (3), we additionally include $H_i$ and the interaction term $D_i \times H_i$ in Eq (1). $H_i$ is a dummy variable equal to one if student $i$'s performance in the midterm examination was relatively high. That is, we use the nonbinary midterm performance variable in the interaction term in Eq (2), while we use the dummy variable for top-half versus bottom-half in the midterm performance in the interaction term in Eq (3). In Eq (3), we set the threshold between high and low performances on the midterm examination to 60 points, where 60 is the *ex ante* threshold score whether the students pass or fail the course.

As shown in Column (1) in Table 4, the estimated coefficient for the interaction term $D_i \times Y_{Mmci}$ is negative (−0.197) but insignificant. It is not confirmed that the impacts of relative performance information feedback linearly depend on the midterm performance. In contrast, as shown in Column (2), the coefficient for $D_i$ is significantly positive and this magnitude is 5.553. The relative performance information feedback for low-performing students on the

**Table 4. Estimation results: The heterogeneity in the effects of relative performance information feedback.**

| Dependent variables | $Y_F$ | |
|---|---|---|
| | **(1)** Coeff. | **(2)** Coeff. |
| $D$ | 3.686* (2.077) | 5.553** (2.602) |
| $D \times Y_{Mmc}$ | -0.197 (0.134) | |
| $Y_{Mmc}$ | 0.827*** (0.096) | |
| $D \times H$ | | -8.074** (3.550) |
| $H$ | | -2.892 (3.575) |
| $Y_M$ | | 0.875*** (0.115) |
| $Class1$ | -2.640 (3.136) | -1.831 (3.140) |
| $Class2$ | -4.685* (2.641) | -3.709 (2.690) |
| $Class3$ | -4.357 (2.961) | -3.622 (2.991) |
| $Female$ | 1.148 (3.115) | 1.363 (3.085) |
| $Constant$ | 63.950*** (2.396) | 20.831*** (6.137) |
| Obs. | 254 | 254 |
| Adjusted $R^2$ | 0.39 | 0.40 |
| F test $H_0$: all the coefficients except the constant are jointly zero | 19.11*** | 18.66*** |
| F test $H_0$: the sum of the coefficients of $D$ and $D \times H$ = the coefficients of $H$ | | 0.01 |

[1] *,** and*** indicate statistical significance at the 10%, 5% and 1% levels, respectively.

[2] Standard errors in parentheses are adjusted for heterogeneity.

[3] Because we exclude a student whose midterm examination score was revised from the sample, our final sample comprised 254 students.

[4] $Y_{Mmc}$ denotes the midterm examination scores which are centered at the mean of $Y_M$.

midterm examination has a positive impact on their performance. Moreover, while the coefficient for the interaction term $D_i \times H_i$ is significantly negative, we do not reject the null hypothesis that "the sum of the coefficients for $D_i$+ ($D_i \times H_i$) equals the coefficients for $H_i$" in Column (2). This indicates that, as for high-performing students, there is no significant difference between the final examination scores for students who *received* their rank information for the midterm examination and the scores for students who *did not receive* their rank information. That is, relative performance information feedback does not have a significant impact on high-performing students.

In Economics I and II, when student *i* took the midterm examination but not the final examination, we usually set $Y_F$ for student *i* to zero. The official university guidelines also state that a student who does not take the final examination fails the course. In our experiment, 12 students who took the midterm examination did not take the final examination. Table 5 shows the number of students whose final examination score is zero by control and treatment groups. We can see that, while the midterm examination scores for the students in both two groups are lower, the number of students in the control group was greater than the number of students in

**Table 5. The number of students whose final examination scores are zero.**

| Score in the midterm examination | Control | Treatment |
|---|---|---|
| 18 | 2 | 0 |
| 19 | 0 | 1 |
| 28 | 4 | 0 |
| 32 | 1 | 1 |
| 37 | 0 | 1 |
| 38 | 2 | 0 |
| Total | 9 | 3 |

the treatment group. This suggests that relative performance information feedback could prevent the students with lower scores in the midterm examination from dropping out of the final examination.

When we treat the students who did not take the final examination as missing and exclude them from our estimation sample, the coefficients for $D$ in Eqs (1)–(3) are positive but insignificant, as reported in Columns (1)–(4) in Table 6. On the other hand, relative performance information feedback on the midterm examination has a significantly negative impact on dropout of the final examination. Table 7 reports the estimation results using OLS applied to a 0–1 dummy variable $Dropout_i$, which takes the value of one if student $i$ dropped out of the final examination and zero otherwise. Thus, relative performance information feedback prevents low-performing students from dropping out of the final examination.

**Table 6. Robustness check: The effects of relative performance information feedback (excluding students whose final examination scores are zero).**

| Dependent variables | $Y_F$ | $Y_F - Y_M$ | $Y_F$ | $Y_F$ |
|---|---|---|---|---|
| | **(1)** Coeff. | **(2)** Coeff. | **(3)** Coeff. | **(4)** Coeff. |
| $D$ | 0.820 (1.536) | 1.734 (1.792) | 0.958 (1.602) | 2.637 (2.012) |
| $D \times Y_{Mmc}$ | | | -0.074 (0.103) | |
| $Y_{Mmc}$ | | | 0.562*** (0.058) | |
| $D \times H$ | | | | -6.300** (3.121) |
| $H$ | | | | 0.596 (3.356) |
| $Y_M$ | 0.532*** (0.050) | | | 0.572*** (0.091) |
| Obs. | 242 | 242 | 242 | 242 |
| Adjusted $R^2$ | 0.37 | 0.03 | 0.37 | 0.38 |
| F test $H_0$: all the coefficients except the constant are jointly zero | 24.54*** | 2.92** | 22.41*** | 20.15*** |
| F test $H_0$: the sum of the coefficients of $D$ and $D \times H$ = the coefficients of $H$ | | | | 0.86 |

[1] *,** and *** indicate statistical significance at the 10%, 5% and 1% levels, respectively.

[2] Standard errors in parentheses are adjusted for heterogeneity.

[3] Because we exclude students whose final examination score was zero or revised from the sample, our final sample comprised 242 students.

[4] $Y_{Mmc}$ denotes the midterm examination scores which are centered at the mean of $Y_M$.

[5] Coefficients of *Class*1, *Class*2, *Class*3, *Female* and *Constant* are not reported.

**Table 7. Robustness check: The effects of relative performance information feedback on dropout (linear probability model).**

| Dependent variables | *Dropout* | *Dropout* |
|---|---|---|
| | (1)<br>Coeff. | (2)<br>Coeff. |
| $D$ | -0.050*<br>(0.026) | -0.052*<br>(0.027) |
| $D \times Y_{Mmc}$ | | 0.<br>(0.002) |
| $Y_M$ | -0.004***<br>(0.001) | |
| $Y_{Mmc}$ | | -0.005***<br>(0.002) |
| Obs. | 254 | 254 |
| Adjusted $R^2$ | 0.09 | 0.09 |
| F test $H_0$: all the coefficients except the constant are jointly zero | 2.24** | 1.94* |

1) *,** and*** indicate statistical significance at the 10%, 5% and 1% levels, respectively.

2) Standard errors in parentheses are adjusted for heterogeneity.

3) Because we exclude a student whose midterm examination score was revised from the sample, our final sample comprised 254 students.

4) $Y_{Mmc}$ denotes the midterm examination scores which are centered at the mean of $Y_M$.

5) Coefficients of *Class*1, *Class*2, *Class*3, *Female* and *Constant* are not reported.

In our experiment, we confirm the following key finding: relative performance information feedback indeed exploits the incentive to study of low-performing students in the midterm examination, but it has no significant impact on high-performing students.

## Discussion

We demonstrate that the significant positive impact of relative performance information feedback is mainly caused by the impact on low-performing students rather than that on high-performing students. What is the reason for this? Suppose that a student aspires to achieve a higher score than another student. The relative performance information feedback tells her the distribution of abilities of students, and how much effort to put in to beat her competitor. Thus, a student with feedback faces the decreased uncertainty of the threshold needed to pass. However, the impact of this information feedback may be asymmetric due to the relative position of the student to her rival. That is, if the student knows that she is already ahead of her rival, she may slack off; if the student knows that she is tied with her rival, she would do her best; if the student knows that she is behind her rival, she may give up. Therefore, relative performance information feedback may potentially diminish the incentive to study in students with very low midterm examination scores even in our setting.

However, we did not find such detrimental effects. Instead, we find that relative performance information feedback reduces the number of students who drop out before the final examination. On this basis, our result is closely related to the effect of class size on students' incentive to study, as demonstrated by Andreoni and Brownback [3] and Brownback [16]. These studies relate an increase in the number of enrollments in an all-pay auction to a decrease in the uncertainty of the threshold needed to pass in a tournament. Their theoretical model predicts that there are aggregately positive but heterogeneously mixed impacts; a

decreasing degree of uncertainty as measured by an increasing class size elicits the effort of high-ability students but suppresses the effort of low-ability students. Andreoni and Brownback [3] confirm this theoretical prediction in a laboratory experiment. In contrast, Brownback [16] conducts an actual classroom experiment, and finds that although increasing class size has positive aggregate impacts, there is no negative impact on the effort of low-ability students. This is similar to our findings.

Some other field experiments also demonstrate that treatments regarding relative rewarding or relative performance information feedback have negative impacts on the students with relatively low abilities. Campos-Mercade and Wengstrom [17] conduct field experiments in an actual university program to examine how monetary reward affects the threshold incentive. In their experiments, students in the treatment group received a scholarship if they reached a certain GPA. The authors demonstrate that only the treatment effects on the male students who are just below the thresholds are significantly positive in the short run. This result is also consistent with ours. However, Campos-Mercade and Wengstrom [17] also demonstrate that the treatment effects on the female students whose initial abilities are low are significantly negative in the long run. Bedard and Fischer [18] examine the effect of relative evaluation on students' examination performance using a field experiment in an actual university classroom. They find that the relative evaluation scheme negatively affects the performance of students who consider they are relatively low in the ability distribution. Ashraf et al. [11] conduct a field experiment in a health assistant training program in Zambia. In their experiment, student rewards are absolute, with some students advised that they will receive a rank-related reward. The authors conclude that the performance of students whose initial achievement level is relatively low is significantly lower when it is announced that they will receive a rank-related reward. These results suggest that relative performance information feedback might have potentially detrimental impacts on students with low ability.

In our experiment, however, information feedback prevents students with low midterm exam scores from dropping out. This may be due to a number of factors that are specific to our experiment. For example, the midterm examination score distribution may be sufficiently converged for students. If the ability distribution is more diverged, relative performance information feedback may have a detrimental impact even in our experimental framework. Other factors may also prevent students with low midterm scores from dropping out. Further research is required to identify the determinants of the incentive to study for low-ability students.

Finally, we should note a limitation concerning the generalizability of our results. Once we consider the situation in which students' primary concern is to obtain the highest possible grade, the results may be altered. That is, if there are multiple thresholds that students would like to pass, relative performance information feedback even has a significant positive impact on high-performing students. Indeed, this may be the case in the study by Tran and Zeckhauzer [10]. In their experiment, rank has no tangible benefit, but students may have an inherent preference for rank. Gill et al. [19] conduct a laboratory experiment to examine the impact of relative performance information feedback on the subjects' performance when rewards are absolute, that is, where rewards are independent of the other subjects' performance. They find that the rank response function is U-shaped, that is, subjects increase their effort the most in response to relative performance information feedback when they are ranked first or last. The authors argue that the U-shaped response function is caused by the combination of pride or "joy of winning" from achieving a high rank together with an aversion to a low rank. Further research is needed to identify the most effective grading and information feedback scheme that can elicit the incentive to study for high-ability students.

## Conclusions

Our experimental results demonstrate that relative performance information feedback has a positive impact on a student's examination score in a relative grading environment where she takes examinations multiple times. In particular, this positive impact is indeed significant for low-performing students in the previous examination. As emphasized in Becker and Rosen [1], a student's position in the distribution of academic attainment is crucial in relative grading.

There are two important areas for future research: one is to identify the determinants of the incentive to study for low-performing students; the second is to identify the most effective grading and information feedback scheme for eliciting the incentive to study for high-performing students.

## Supporting information

**S1 File. Steps to replicate the tables and figures.** The instruction to replicate the tables and figures in "Information Feedback in Relative Grading: Evidence from a Field Experiment" by Shinya Kajitani, Keiichi Morimoto and Shiba Suzuki.
(PDF)

**S2 File. Stata program file.**
(DO)

**S1 Dataset. This is the CSV file that contains our datasets.**
(CSV)

## Acknowledgments

We are very grateful to four anonymous reviewers of this journal, Akira Yamazaki, Kentaro Kobayashi, Hayato Nakata, and Masahiro Watabe for invaluable comments and suggestions. We also thank Naohito Abe, Kosuke Aoki, David Gill, Tao Gu, Shigeki Kano, Vu Tuan Kai, Nobuyoshi Kikuchi, Jun-Hyung Ko, Colin McKenzie, Akira Miyaoka, Koyo Miyoshi, Tomoharu Mori, Chang-Min Lee, Masao Nagatsuka, Kengo Nutahara, Fumio Ohtake, Daniela Puzzello, David Reiley, Masaru Sasaki, Michio Suzuki, Kan Takeuchi, Ryuichi Tanaka, Tomoaki Yamada and participants at annual meeting on Japanese Economic Association 2013 Spring Meeting (Toyama, Japan), Economic Science Association European meeting 2015 (Heidelberg, Germany), Asian and Australasian Society of Labour Economics 2018 Conference (Seoul National University), Kansai Labor Workshop, and seminar participants at Meiji University, Meisei University, Seikei University, and the University of Tokyo for their helpful comments.

## Author Contributions

**Conceptualization:** Shinya Kajitani, Keiichi Morimoto, Shiba Suzuki.

**Data curation:** Shinya Kajitani.

**Formal analysis:** Shinya Kajitani.

**Investigation:** Shinya Kajitani, Shiba Suzuki.

**Methodology:** Keiichi Morimoto, Shiba Suzuki.

**Project administration:** Shinya Kajitani, Keiichi Morimoto.

**Writing – original draft:** Shinya Kajitani, Keiichi Morimoto, Shiba Suzuki.

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
