## [Decision Letter · Decision Letter 0]

10 Feb 2020

PONE-D-19-35238

Information feedback in relative grading: Evidence from a field experiment

PLOS ONE

Dear Prof. Suzuki,

Thank you for submitting your manuscript to PLOS ONE. After careful consideration, we feel that it has merit but does not fully meet PLOS ONE’s publication criteria as it currently stands. Therefore, we invite you to submit a revised version of the manuscript that addresses the points raised during the review process.

Though Reviewer 4 rejected PONE-D-19-35238, the reviewer provided many valuable and constructive comments. Considering reviewers’ useful comments and the interesting topic of the manuscript, I would like to give you a chance to revise your manuscript. The revised manuscript will undergo the next round of review by the same reviewers.

We would appreciate receiving your revised manuscript by Mar 26 2020 11:59PM. To enhance the reproducibility of your results, we recommend that if applicable you deposit your laboratory protocols in protocols.io, where a protocol can be assigned its own identifier (DOI) such that it can be cited independently in the future. For instructions see: http://journals.plos.org/plosone/s/submission-guidelines#loc-laboratory-protocols

We look forward to receiving your revised manuscript.

Kind regards,

Baogui Xin, Ph.D.

Academic Editor

PLOS ONE

Journal Requirements:

Reviewers' comments:

Reviewer's Responses to Questions

**Comments to the Author**

1. Is the manuscript technically sound, and do the data support the conclusions?

Reviewer #1: Partly

Reviewer #2: Yes

Reviewer #3: Yes

Reviewer #4: Partly

2. Has the statistical analysis been performed appropriately and rigorously? 

Reviewer #1: Yes

Reviewer #2: Yes

Reviewer #3: Yes

Reviewer #4: No

3. Have the authors made all data underlying the findings in their manuscript fully available?

Reviewer #1: Yes

Reviewer #2: Yes

Reviewer #3: Yes

Reviewer #4: Yes

4. Is the manuscript presented in an intelligible fashion and written in standard English?

Reviewer #1: Yes

Reviewer #2: Yes

Reviewer #3: Yes

Reviewer #4: Yes

5. Review Comments to the Author

Reviewer #1: The authors conducted a randomized controlled trial at a Japanese university to examine the impact of relative performance information feedback of students' performance on their examination score under the relative. Their results show that relative performance information feedback has a significantly positive impact on the students' final examination score on average, but that the average positive impact is derived by the improvement of low-performing students.

In general, this is not a large sample for RCT, but I think it’s an interesting topic and authors perform a good study design and data analysis. However, I fear that this will require quite some work with respect to the paper.

First, although authors claim that it would be generally difficult for a student in the control group to identify a student in the treatment group with exactly the same score and it leads to the ITT effects may underestimate the average causal effects on those in fact treated, my major concern still the information spill over that students in both the control and treatment groups sit in the same classroom can exchange their information from letters. Actually, it is easy for students to exchange their score and rank information in the same classroom, especially in the classroom 1 (only 40 students), and students may guess why only a part of students receive rank information that would disturb the treatment effect. Authors should make more clarification or verification about this problem.

Second, authors should report more details on experiment manipulation. For example, how they avoid different pass standards for exam scores from different instructors? Did students pass exam depend on specific distribution of all the scores, and whether students know it? Whether there was a strict demand from school on distribution of every score level (A/B/C/D) for instructors?

Third, authors can do more work on data analysis. The paper presents some reasons that authors do not report robust standard errors on classroom level. I want to know whether they can try to cluster at the other level, such as dormitory. The paper does not report the variables they controlled in the regression. In addition, whether they can try to use the difference between the midterm exam score and the final exam score as the explained variable.

Reviewer #2: Review overview:

In this paper, the authors look at how effort responds to information about relative position in a classroom graded on the curve. The primary treatment arm randomly assigns students to receive information about their midterm exam grade relative to the population of students. Then, the authors measure how this affects performance on the final exam. As a result of this information intervention, higher ability students reduce their motivated relative to lower ability students.

This paper has chosen an important topic and an ambitious approach to studying it. I have no concerns about this being of broad interest.

Please see attached file for the remainder of the review.

Reviewer #3: Report on PONE-D-19-35238: Information feedback in relative grading: Evidence from a field experiment by Shinya Kajitani, Keiichi Morimoto, & Shiba Suzuki

Summary: The authors report the results from a field experiment with economics students testing the impact of providing relative performance feedback in an intermediate exam on students' performance in the final exam. They find that the intermediate relative performance feedback has a significantly positive effect on students' performance in the final exam on average, and that this effect is mainly driven by students who performed badly in the intermediate exam.

Comments:

This paper reports interesting results from a well-conducted field experiment in a relevant setting with real incentives. I am in general quite positive about the paper, however, I do have some comments and questions that the authors should or could address in a revision.

Major comments

1) Generalizability of the result that relative performance feedback mainly affects low performers: One of the main results of the experiment, which the authors mention already in the abstract, is that the relative performance feedback had an effect mainly by increasing the performance of students who performed badly in the intermediate exam. While this effect is interesting and is supported by the data analysis of the current experiment, it seems important to state that this effect cannot simply be generalized to other settings in which relative performance feedback is given. From a theoretical perspective, on which groups relative performance feedback has an effect likely depends on the distribution of performance or skills differences in the population (and, of course, people's knowledge about or perception of these distributions) and on the exact incentive structure. If the incentive structure is such that only very top performers are rewarded or if initial low performers have much lower skills (higher effort costs) than their higher-skilled competitors, then it may well be the case that relative performance feedback actually demotivates low performers and has a more motivating effect on people in the middle of the skill / performance distribution. Thus, while I believe the result the authors find in the current setting is interesting, I think it should be made clear that one cannot automatically expect it to generalize to other settings where relative performance feedback is used.

2) Observations of students who did not take the final exam: Currently the authors set the scores of students who did not take the final exam to zero. This seems to be a strong assumption that is not necessarily warranted. Another approach would be to simply treat these observations as missing and exclude them from the analyses. As a robustness check, I would like to see the results of the regression analyses if this way of dealing with these cases is used. Moreover, I find the authors' use of the "Intention to Treat" (ITT) effect in the paper confusing. To me an ITT effect would much rather be including those students who were randomized into a certain group but did not receive the treatment (i.e., the feedback).

3) Estimators used in regression analyses: what estimator was used in the regressions? I assume the authors used a Tobit-estimation, as they speak about left-censoring. If this is the case, I doubt whether it is necessary. It seems perfectly fine to simply use OLS to estimate the effects of treatment interventions on test scores with a lower bound at zero (see, e.g., Levitt et al., 2016). Using OLS would also allow dropping the marginal effects columns, as the coefficients can be directly interpreted. In any case, the authors should be clear about what estimator they used and why.

Minor comments

4) Information / knowledge about incentive structure: The authors write that the "students in Economics II knew the grading scheme … because the instructors already explained the grading scheme in detail at the beginning of the second semester and this grading scheme had already been employed in Economics I" (p. 8). Given that students' knowledge about the relative performance incentive structure created by the grading scheme is a key element of the experiment and the paper, I think it would nice to provide more details on the exact information that students received. Could, for instance, slides or other materials that were used to explain the grading scheme to students be shown in an Appendix? Or, even better, do the authors have any data on students' understanding of the grading system and the incentive structure it creates? The system seems relatively complicated with quite some discretion on the part of the graders, so it is not completely obvious that the students would have understood how the system works exactly and what incentives it creates. Any additional evidence that can be provided in this regard would therefore be helpful and make the interpretation of the experimental results more convincing.

5) Were the graders blind to the treatment a certain student had been assigned to? Ideally, they would have been. In any case, this information should be added.

6) If available, please provide some more information on demographics for the randomization checks in Table 1 (e.g., gender or any other available data).

7) I think the paragraph on how randomization avoids pitfalls of regression to the mean (p. 10) can be deleted. This point is obvious to anybody who has understood how randomization works, and the paragraph does little more than divert the reader's attention.

8) I don't understand how Table 5 allows addressing the point that "relative performance information feedback could positively affect high-performing students more than low-performing students, even if the rank for the high-performing students has less tangible benefits" (p. 15). The only thing that the regressions reported in Table 5 do differently compared to those in Table 4 is to use a non-binary intermediate performance variable (the score in the intermediate exam) in the interaction term. This is a relevant robustness check, as the dummy for top-half vs. bottom-half in the intermediate exams contains less information than the quasi-continuous score variable. I would suggest that the authors motivate Table 5 that way, or alternatively, explain better how this analysis addresses the point quoted above.

9) Moreover, the authors could consider mean-centering Y_M (the score in the intermediate exam) in the regressions reported in Table 5. Without mean-centering (as is currently the case) the effect of the treatment Dummy D is estimated at Y_M = 0, which is a very special case.

10) Wording / language:

P. 2: "efficient way of eliciting the incentives": "eliciting" doesn't seem to be the right word to me here. Maybe "increasing" would be better?

P. 3: First sentence of the second paragraph (starting with "revealing the role of…") needs to be rewritten.

P. 6: First sentence in second full paragraph: "The randomized controlled trial was conducted immediately after…": I think it would be more appropriate to write something like "The experimental intervention was implemented immediately after…."

P. 9: I would suggest calling Table 1 "Randomization Checks" instead of "Confirmation of randomness"

Congratulations to the authors on a very nice paper and all the best for their future work!

References:

Levitt, S. D., List, J. A., Neckermann, S., & Sadoff, S. (2016). The behavioralist goes to school: Leveraging behavioral economics to improve educational performance. American Economic Journal: Economic Policy, 8(4), 183-219.

Reviewer #4: This paper reports results of an RCT study on the effect of relative performance information feedback on students’ examination score in a compulsory subject in a Japanese university. Authors find a significant positive effect but the average treatment effect is driven by improvement of scores of low-performing students but not high-performing students. Though there are a few studies on effect of relative grading feedback on scores, there is a dearth of studies on effect of relative grading feedback on scores in a real educational environment. In this context, the study contributes to vast empirical literature on grade incentives. However, there are major issues that need to be addressed. Please find below my specific and general comments.

Specific Comments:

1. The design of experiment has not been explained clearly. For instance, what was the logic behind having Classroom 1 with higher ability students being selected into it? Is it because they are deemed to be more motivated than others or the stratification was driven by other aspects? What were the variations for the three other classrooms? Was there a neutral framing in the group assignment?

2. Any insights on the power calculations would have been useful to understand if the cell sizes are statistically justified?

3. The experiment procedure suggests several threats to internal validity. Of particular concern are the spill over effects. The authors have raised concerns about both treatment and control groups sitting in the same classroom. Was there any control over communications among students about the different grading patterns?

4. How does the treatment result in the observed impact? There is a debate on how relative grading influences motivation that has not been discussed. Aspects of direct competition among peers is not evident.

5. Does the relative performance feedback affect self- perceived ability, cognitive tactics, strategies, reinforcing cues, or identity rather than directly affecting effort? How does the effort incentives differ for those with low and high grades?

6. Theoretically relevant interaction effects and robustness checks could be examined. E.g. past academic history and relative grading.

7. Are there gender gaps in performance such that the sex-ratios in classrooms need to be controlled for?

8. Role of teachers is not clear in this experiment. There is no information on teacher ability differences that may impact students’ scores. This is particularly disconcerting because feedback and instructions have interactions that have been shown to impact performance.

General Comments:

1. The paper is generally well written but there are a few typos that can be addressed.

2. The review of literature can improve, and major theoretical aspects relevant to the paper can be developed.

3. Please verify the claim “By contrast, no existing study examines the impact of relative performance information feedback on student incentives under relative grading in an actual educational environment.”

6. PLOS authors have the option to publish the peer review history of their article (what does this mean?). If published, this will include your full peer review and any attached files.

Reviewer #1: Yes: Jun Luo

Reviewer #2: No

Reviewer #3: Yes: Manuel Grieder

Reviewer #4: No

---

## [Author Response · Author response to Decision Letter 0]

24 Mar 2020

Responses to Editor and Reviewers

Dear Professor Baogui Xin and Reviewers,

Thank you very much for giving us an opportunity to submit a revised version of our manuscript entitled “Information feedback in relative grading: Evidence from a field experiment” (Manuscript ID: PONE-D-19-35238). We would like to thank the referees for their insightful comments, all of which have contributed to improving our manuscript. We have followed the referees’ comments and we summarize our responses to their comments below. The major revised portions are marked in red in the revised manuscript. Our detailed responses to all of the comments are in “Reply_Letter.pdf”. 

 We hope that this version of our paper is now deemed suitable for publication and we look forward to hearing from you in due course.

Sincerely, 

Shiba Suzuki, PhD

Professor of Economics, Seikei University

 

Responses to reviewers’ comments

Reviewer #1

1.1) First, although authors claim that it would be generally difficult for a student in the control group to identify a student in the treatment group with exactly the same score and it leads to the ITT effects may underestimate the average causal effects on those in fact treated, my major concern still the information spill over that students in both the control and treatment groups sit in the same classroom can exchange their information from letters. Actually, it is easy for students to exchange their score and rank information in the same classroom, especially in the classroom 1 (only 40 students), and students may guess why only a part of students receive rank information that would disturb the treatment effect. Authors should make more clarification or verification about this problem.

In response to this comment, we confirm that most of the students in the control group cannot know their own rank exactly even if students in Classroom 1 aggregate the rank information they received. Thus, the degree of uncertainty that students in the control group face is much higher than that for students in the treatment group. This point is now added as a separate paragraph in lines 271–288 in the revised paper.

1.2) Second, authors should report more details on experiment manipulation. For example, how they avoid different pass standards for exam scores from different instructors? Did students pass exam depend on specific distribution of all the scores, and whether students know it? Whether there was a strict demand from school on distribution of every score level (A/B/C/D) for instructors?

In response to this comment, the slide that was used to explain the grading criteria in the guidance for freshman students in April is added as Fig. 4 in the revised paper (p. 8). In addition, we also explain that instructors verbally announced the possibility of raw scores being adjusted upward (lines 185–190 in the revised paper). There was no strict demand from school on distribution of every score level for instructors. 

 In addition, we have added an explanation (lines 219–227) in the revised paper to stress that each instructor could not deviate from the agreed pass scores.

1.3) Third, authors can do more work on data analysis. The paper presents some reasons that authors do not report robust standard errors on classroom level. I want to know whether they can try to cluster at the other level, such as dormitory. The paper does not report the variables they controlled in the regression. In addition, whether they can try to use the difference between the midterm exam score and the final exam score as the explained variable.

There is no cluster at the other level (such as dormitory) in our experimental environment. However, we did not adjust the standard errors for individual heterogeneity. We have rewritten the text on this issue (lines 305–307) and show the robust standard errors in all tables in the revised paper. 

 The variables we controlled in the regression were D, YM, Class1, Class2, and Class3, as shown in Table 3 in the previous version (p. 13). In the revised paper, we additionally control for the sex dummy Female in the regression. We highlight this in lines 302–304 in the revised paper. 

 Moreover, as the reviewer suggested, we include the difference between the midterm examination score and the final examination score (YF − YM) instead of YM as the independent variable. As shown in Table 3 in the revised paper (p. 14), the magnitude of the coefficient of D in Column (2) (3.706) is similar to that in Column (1) (3.517). We point this out in lines 315–319 in the revised paper.

Reviewer #2: 

2.1) By separating the top 40 students, the interpretation of information feedback becomes challenging. Moreover, the treatment assignment weights are different between classroom 1 and classrooms 2-4. 

- I’m a bit confused about Table 3, where there are class-level indicator variables. In this case, dropping Class 1 should only change the precision of the estimates of other covariates, but shouldn’t change the estimation of the treatment effects, because there is no within-class-1 variation in treatment. So, I think you could even drop columns (2a) and (2b) and say it’s taken care of by the indicator variables. 

- When you think about the treatment on the treated, it is also challenging to think about how this classroom affects that. If I’m not in Class 1, then I already know something about my relative score, so it’s a little strange. I would like to see this explained a bit in the text. 

Thank you for your insightful suggestion that “dropping Classroom 1 should only change the precision of the estimates of other covariates, but should not change the estimation of the treatment effects.” We have now deleted the estimation results shown in Columns (2a) and (2b) in Tables 3–5 in the previous paper. 

2.2) The fact that the courses are not explicitly graded on a curve is a little strange. The curving is more important for the lower performing students because it can only move scores upwards. I would like to see the paper mention how an asymmetric curve may affect lower ability students differently from higher ability ones. 

A few papers that I would cite are:

- Gill, David, et al. "First-place loving and last-place loathing: How rank in the distribution of performance affects effort provision." Management Science 65.2 (2019): 494-507.

-This could help inform the discussion of how rankings matter for students.

-Bedard, Kelly, and Stefanie Fischer. "Does the response to competition depend on perceived ability? Evidence from a classroom experiment." Journal of Economic Behavior & Organization 159 (2019): 146-166.

- "Threshold Incentives and Academic Performance," Pol Campos-Mercade and Erik Wengström

- Brownback, Andy. "A classroom experiment on effort allocation under relative grading." Economics of Education Review 62 (2018): 113-128.

These three can address the existing literature on how perceptions or realities about relative ranking affect effort choices. 

We appreciate this comment because these suggested papers have substantially improved our manuscript. In response, we added the new subsection “Discussion” (pages 19-22) in the revised paper to relate our experimental results to those of existing research. In particular, the first and the second paragraphs in the subsection “Discussion” (p. 19) state the possible asymmetric impact of relative performance information feedback. 

Reviewer #3: 

3.1) Generalizability of the result that relative performance feedback mainly affects low performers: One of the main results of the experiment, which the authors mention already in the abstract, is that the relative performance feedback had an effect mainly by increasing the performance of students who performed badly in the intermediate exam. While this effect is interesting and is supported by the data analysis of the current experiment, it seems important to state that this effect cannot simply be generalized to other settings in which relative performance feedback is given. From a theoretical perspective, on which groups relative performance feedback has an effect likely depends on the distribution of performance or skills differences in the population (and, of course, people's knowledge about or perception of these distributions) and on the exact incentive structure. If the incentive structure is such that only very top performers are rewarded or if initial low performers have much lower skills (higher effort costs) than their higher-skilled competitors, then it may well be the case that relative performance feedback actually demotivates low performers and has a more motivating effect on people in the middle of the skill / performance distribution. Thus, while I believe the result the authors find in the current setting is interesting, I think it should be made clear that one cannot automatically expect it to generalize to other settings where relative performance feedback is used.

We really appreciate this comment. In response, we have added the new subsection “Discussion” (pages 19-22) in the revised paper to discuss the generalizability of our results. In particular, we relate our experimental results to those of existing research that stress the importance of students’ ability distribution. For example, the paragraphs starting line 443 and 464 state the possible negative impact of relative performance information feedback on students with low midterm scores. The paragraph starting line 472 suggests the possible positive impact on students with high midterm scores. 

3.2) Observations of students who did not take the final exam: Currently the authors set the scores of students who did not take the final exam to zero. This seems to be a strong assumption that is not necessarily warranted. Another approach would be to simply treat these observations as missing and exclude them from the analyses. As a robustness check, I would like to see the results of the regression analyses if this way of dealing with these cases is used. Moreover, I find the authors' use of the "Intention to Treat" (ITT) effect in the paper confusing. To me an ITT effect would much rather be including those students who were randomized into a certain group but did not receive the treatment (i.e., the feedback).

This is an insightful suggestion for revising our paper. In both Economics I and II, when students did not take the examination, we usually set their examination score to zero. In Table 5 in the revised paper, we can see that, while the midterm examination scores for students whose final examination score is zero in both the control and treatment groups are lower, the number of students in the control group who did not take the final examination was greater than the number of students in the treatment group. This suggests that relative performance information feedback could prevent the students with lower scores in the midterm examination from dropping out of the final examination. When we exclude them from our estimation sample, the coefficients for D are positive but insignificant, as reported in Columns (1)–(4) in Table 6 in the revised paper. On the other hand, relative performance information feedback on the midterm examination has a significantly negative impact on dropout of the final examination Dropout, as shown in Table 7 in the revised paper. We have suggestive evidence that relative performance information feedback prevents low-performing students from dropping out of the final examination. We point out these issues in the revised paper (lines 391-401 and 402-410). Moreover, as the reviewer rightly notes, it was inappropriate to use the “intention-to-treat” (ITT) effect in the earlier version of the paper. We have rewritten the relevant text in the revised paper.

3.3) Estimators used in regression analyses: what estimator was used in the regressions? I assume the authors used a Tobit-estimation, as they speak about left-censoring. If this is the case, I doubt whether it is necessary. It seems perfectly fine to simply use OLS to estimate the effects of treatment interventions on test scores with a lower bound at zero (see, e.g., Levitt et al., 2016). Using OLS would also allow dropping the marginal effects columns, as the coefficients can be directly interpreted. In any case, the authors should be clear about what estimator they used and why.

We have rewritten the estimation model (first paragraph in the section “Results and Discussion” on p. 12 of the revised paper) and show the revised estimation results in Column (1) in Table 3 in the revised paper.

3.4) Information / knowledge about incentive structure: The authors write that the "students in Economics II knew the grading scheme … because the instructors already explained the grading scheme in detail at the beginning of the second semester and this grading scheme had already been employed in Economics I" (p. 8). Given that students' knowledge about the relative performance incentive structure created by the grading scheme is a key element of the experiment and the paper, I think it would nice to provide more details on the exact information that students received. Could, for instance, slides or other materials that were used to explain the grading scheme to students be shown in an Appendix? Or, even better, do the authors have any data on students' understanding of the grading system and the incentive structure it creates? The system seems relatively complicated with quite some discretion on the part of the graders, so it is not completely obvious that the students would have understood how the system works exactly and what incentives it creates. Any additional evidence that can be provided in this regard would therefore be helpful and make the interpretation of the experimental results more convincing.

In response to this comment, we have added the slide to explain the grading scheme for freshman students in April as Fig. 4 on p. 9. Although the slide only states the relationship between scores and grades, we also verbally explained that there is a possibility that raw scores are adjusted upward to obtain a reasonable pass rate. In addition, we explain the grading scheme of Economics I in more detail in lines 201–217. 

3.5) Were the graders blind to the treatment a certain student had been assigned to? Ideally, they would have been. In any case, this information should be added.

In response to this comment, we have explained the grading scheme of Economics I in more detail, and the following text has been added (lines 146–153). 

3.6) If available, please provide some more information on demographics for the randomization checks in Table 1 (e.g., gender or any other available data).

We additionally provide a randomization check (Table 1 in the revised paper). There is no significant difference in the mean scores between the control and treatment groups by sex, as clarified in lines 269–270. 

3.7) I think the paragraph on how randomization avoids pitfalls of regression to the mean (p. 10) can be deleted. This point is obvious to anybody who has understood how randomization works, and the paragraph does little more than divert the reader's attention.

As the reviewer suggested, we have deleted this point in the revised paper.

3.8) I don't understand how Table 5 allows addressing the point that "relative performance information feedback could positively affect high-performing students more than low-performing students, even if the rank for the high-performing students has less tangible benefits" (p. 15). The only thing that the regressions reported in Table 5 do differently compared to those in Table 4 is to use a non-binary intermediate performance variable (the score in the intermediate exam) in the interaction term. This is a relevant robustness check, as the dummy for top-half vs. bottom-half in the intermediate exams contains less information than the quasi-continuous score variable. I would suggest that the authors motivate Table 5 that way, or alternatively, explain better how this analysis addresses the point quoted above.

We reported the estimation results shown in Table 5 in the previous paper as the relevant robustness check for the estimation results shown in Table 4 in the previous paper. We have rewritten this information (lines 365–390) in the revised paper.

3.9) Moreover, the authors could consider mean-centering Y_M (the score in the intermediate exam) in the regressions reported in Table 5. Without mean-centering (as is currently the case) the effect of the treatment Dummy D is estimated at Y_M = 0, which is a very special case.

We now include the mean-centering YM instead of YM as the independent variable. As shown in Table 4 in the revised paper (p. 14), the magnitude of the coefficient of D in Column (1) (3.686) is similar to that in Column (1) in Table 3 in the revised paper (3.517). We discuss this in lines 365–390 in the revised paper.

3.10) Wording / language:

P. 2: "efficient way of eliciting the incentives": "eliciting" doesn't seem to be the right word to me here. Maybe "increasing" would be better?

P. 3: First sentence of the second paragraph (starting with "revealing the role of…") needs to be rewritten.

P. 6: First sentence in second full paragraph: "The randomized controlled trial was conducted immediately after…": I think it would be more appropriate to write something like "The experimental intervention was implemented immediately after…."

P. 9: I would suggest calling Table 1 "Randomization Checks" instead of "Confirmation of randomness"

As the reviewer suggested, we have corrected all sentences.

Reviewer #4: 

4.1) The design of experiment has not been explained clearly. For instance, what was the logic behind having Classroom 1 with higher ability students being selected into it? Is it because they are deemed to be more motivated than others or the stratification was driven by other aspects? What were the variations for the three other classrooms? Was there a neutral framing in the group assignment?

We have rewritten the design of our experiment in the last paragraph on p. 7 in the revised paper. In particular, we have added the reason why we placed students with a top-40 score in the Pretest of Mathematics in Classroom 1. (lines 154-170)

4.2) Any insights on the power calculations would have been useful to understand if the cell sizes are statistically justified?

This is a very insightful suggestion and we mention this issue in footnote 2 in Table 2 in the revised paper.

4.3) The experiment procedure suggests several threats to internal validity. Of particular concern are the spill over effects. The authors have raised concerns about both treatment and control groups sitting in the same classroom. Was there any control over communications among students about the different grading patterns?

In response to this comment, we confirm that most of students in the control group cannot know their own rank exactly even if students in Classroom 1 aggregate the rank information they received. Thus, the degree of uncertainty that students in the control group face is much higher than for students in the treatment group. This point has been added as a separate paragraph (lines 271–288) in the revised paper. 

4.4) How does the treatment result in the observed impact? There is a debate on how relative grading influences motivation that has not been discussed. Aspects of direct competition among peers is not evident.

As the reviewer pointed out, the relative grading scheme is itself an important issue and it is worth considering further. A recent paper (Czibor et al. [15]) reported that relative grading could not elicit the incentive to study compared with absolute grading when students are less interested in achieving a higher grade. As our experimental framework is close to theirs, we have added some explanation (lines 334–350) in the revised version. 

4.5) Does the relative performance feedback affect self- perceived ability, cognitive tactics, strategies, reinforcing cues, or identity rather than directly affecting effort? How does the effort incentives differ for those with low and high grades?

We really appreciate this comment. In response, we have added the new subsection “Discussion” (pages 19-22) in the revised paper to discuss the issues raised. In particular, we relate our experimental results to those of existing research that argue for the importance of students’ ability distribution. For example, in the paragraphs starting line 443 and 464, we note the possible negative impact of relative performance information feedback on students with low midterm scores; the paragraph starting line 472 suggests the possible positive impact on students with high midterm scores. 

4.6) Theoretically relevant interaction effects and robustness checks could be examined. E.g. past academic history and relative grading.

We demonstrate that the significant positive impact of relative performance information feedback is mainly caused by the impact on low-performing students rather than that on high-performing students. We discuss these results from a theoretical perspective in the subsection “Discussion” (pages 19-22) in the revised paper.

In response to the reviewer’s suggestion, we examine whether the effect of relative performance information feedback depends not on the distribution of performance in the midterm examination (the midterm examination score Y_M) but on students’ skills differences, which are assessed from their past academic history: the final score in Economics I Econ1 and the score in the Pretest of Mathematics Math. We use Econ1 and Math as the interaction terms with D instead of Y_M:

Y_Fi=α_1 D_i+α_2 (D_i×H_Econ1i )+α_3 H_Econ1i+X_i β_1+u_1i (A)

Y_Fi=α_4 D_i+α_5 (D_i×H_Mathi )+α_6 H_Mathi+X_i β_2+u_2i (B)

where H_Econ1i in Equation (A) denotes a dummy variable equal to one if student i’s performance in Economics I was relatively high (above average score), and H_Mathi in Equation (B) denotes a dummy variable equal to one if student i’s performance in the Pretest of Mathematics was relatively high (above average score). 

 Both the coefficients α_2 and α_5 are insignificant. The results suggest that, compared with the results shown in Column (2) in Table 4 in the revised paper, the effect of relative performance information feedback depends on the distribution of performance in the midterm examination per se. These results are not surprising once we consider students’ incentive to study. Because students have to pass the course, the score of the midterm examination is important for them. We believe that this analysis is very helpful in interpreting our results. However, due to time and space constraints, we did not include these results in the revised manuscript. 

4.7) Are there gender gaps in performance such that the sex-ratios in classrooms need to be controlled for?

Following this suggestion, we have additionally used the female dummy Female in the estimation model in the revised paper. The coefficients of Female in all the estimation models are insignificant.　We additionally provide a randomization check (Table 1 in the revised paper). There is no significant difference in the mean scores between the control and treatment groups by sex. We have added this information in lines 302–304.

4.8) Role of teachers is not clear in this experiment. There is no information on teacher ability differences that may impact students’ scores. This is particularly disconcerting because feedback and instructions have interactions that have been shown to impact performance.

This is an insightful suggestion. In Table 3 in the revised paper, the coefficient of Class2 in Column (1) and those of Class1–Class3 in Column (2) are significant. The classroom fixed effects would absorb instructor fixed effects as well as peer effects and other classroom-level factors. We point out that these effects and factors may be associated with test scores in lines 319–322 in the revised paper.

In addition, it is noted that instructors were not technically blind regarding which students had been assigned to the treatment group. However, apart from handing letters to students, instructors could not confirm which students were assigned to treatment or control, and it would be difficult for instructors to remember this information. In addition, class attendance or participation, such as the number of times a student spoke in class, was not evaluated at all. Thus, whether instructors were blind to which students were assigned to treatment or control would have little impact on the experimental results. We address this issue in lines 146–153 in the revised paper.

4.9) The paper is generally well written but there are a few typos that can be addressed.

We have now carefully corrected typos. 

4.10) The review of literature can improve, and major theoretical aspects relevant to the paper can be developed.

In response to this comment, we added the new subsection “Discussion” (pp. 19–22) in the revised paper to relate our experimental results to those of existing research. 

4.11) Please verify the claim “By contrast, no existing study examines the impact of relative performance information feedback on student incentives under relative grading in an actual educational environment.”

To our knowledge, there is no previous study that examined the impact of relative performance information feedback on student incentives under relative grading in an actual educational environment. However, in response to this comment, we have deleted the expression noted.

---

## [Editor Report · Decision Letter 1]

26 Mar 2020

Information feedback in relative grading: Evidence from a field experiment

PONE-D-19-35238R1

Dear Dr. Suzuki,

We are pleased to inform you that your manuscript has been judged scientifically suitable for publication and will be formally accepted for publication once it complies with all outstanding technical requirements.

With kind regards,

Baogui Xin, Ph.D.

Academic Editor

PLOS ONE
---

## [Editor Report · Acceptance letter]

1 Apr 2020

PONE-D-19-35238R1 

Information feedback in relative grading: Evidence from a field experiment 

Dear Dr. Suzuki:

I am pleased to inform you that your manuscript has been deemed suitable for publication in PLOS ONE. Congratulations! Your manuscript is now with our production department. 

With kind regards,

on behalf of

Prof. Baogui Xin 

Academic Editor

PLOS ONE